# Psychometric Properties of the Athletic Shoulder Test in Adolescent Tennis Players

**DOI:** 10.3390/jcm14041146

**Published:** 2025-02-10

**Authors:** Achilleas Paliouras, Marina Porgiopoulou, Giorgos Varverakis, Giorgos Stavrakakis, Nikolaos Strimpakos, Eleni Kapreli

**Affiliations:** 1Clinical Exercise Physiology and Rehabilitation Research Laboratory, Department of Physiotherapy, School of Health Sciences, University of Thessaly, 35132 Lamia, Greece; marinaporgio@gmail.com (M.P.); gvarverakis1@gmail.com (G.V.); giorgosstavrakas@gmail.com (G.S.); ekapreli@uth.gr (E.K.); 2Health Assessment and Quality of Life Research Laboratory, Department of Physiotherapy, School of Health Sciences, University of Thessaly, 35132 Lamia, Greece; nikstrimp@uth.gr; 3Division of Musculoskeletal and Dermatological Sciences, School of Biological Sciences, University of Manchester, Manchester M13 9PL, UK

**Keywords:** athletic shoulder test, dynamometry, tennis, adolescent, reliability

## Abstract

**Background/Objectives**: The Athletic Shoulder Test (ASH) has been described as one of the most promising upper-extremity tests to assess performance in overhead athletes. Its high reliability rates, short testing period, and applicability in any environment with portable and cheap equipment have been highlighted as some of the advantages of the test. However, it has yet to be evaluated in a non-adult athletic population. Therefore, the aim of this study was to investigate the ASH test’s psychometric properties in a sample of young tennis players. **Methods**: A total of 33 adolescent tennis players were evaluated among two sessions with a week interval. Intra-rater, inter-rater, and test-retest reliability were investigated. Additionally, possible correlations with measures of rotational shoulder strength and upper-extremity performance were examined. Two novice physiotherapists performed all the measurements following appropriate training. **Results**: The relative reliability scores, as calculated by intraclass correlation coefficient (ICC) indices, were found to be excellent (ICC = 0.924–0.988). Standard error of measurement and minimal detectable change scores have been estimated per position (SEM = 2.74–7.06 N, MDC = 7.55–19.42N). Test-retest reliability provided slightly higher SEM and MDC scores on average (SEM = 3.33–6.47, MDC = 9.32–18.04) than intra-rater and inter-rater reliability. Associations between ASH and the two tests were found to be moderate to strong (r = 0.584–0.856), with the dominant arm providing higher correlation scores (r = 0.605–0.856) than the non-dominant one (r = 0.584–0.823). Absolute values were collected and are provided for all three upper-limb tests; normalized values were calculated for ASH and rotational strength and peak torque only for the ASH measurements. **Conclusions**: The excellent reliability rates establish the ASH test as a highly recommended testing protocol for adolescent tennis players.

## 1. Introduction

Tennis is one of the most popular sports worldwide, with an increasing number of athletes of different levels and age groups reported to have participated during the last decades [1,2]. The evolution of the sport in recent years has reshaped the landscape in terms of the required skills for competition, with excessive physicality ranking among the top demands of the sport [3,4]. This could be a major concern for adolescent athletes, as the increased tendency of early specialization that has been observed during recent years could have detrimental effects on this population, mainly due to repetitive microtrauma and overuse injuries [2,5]. The concerning numbers previously been reported indicate an incidence of 0.2–1.8 injuries per 1000 h of participation, while among different sports injury rates between 18% and 90% have been identified, with higher-level athletes being more affected [6,7].

A generic overview of the most common injuries in overhead athletes, such as tennis players, reveals that adolescents’ and adults’ injuries do not differ significantly, with the shoulder being affected more commonly than any other joint [8,9]. When looking at the risk factors that predispose adolescent overhead athletes to injuries, shoulder strength is considered to be key, as in the adult population [10,11,12,13,14]. In order to identify these deficits and attempt to prevent possible future injuries, strength testing has been proposed as one of the cornerstones of assessment [15]. The most commonly performed tests, used either in isolation or as part of a test battery, have presented mixed results in terms of psychometric properties, meaning that their clinical application and interpretation should be performed with caution [15,16]. Additionally, despite being used in athletic adolescent populations, the investigation of psychometric properties of most tests has taken place only in adult populations, highlighting a limitation in their spontaneous generalizability among different sports, levels, and ages [16,17,18].

According to the Bern Consensus statement in 2022, one of the recommended tests for the assessment of overhead athletes is the Athletic Shoulder Test (ASH) [15]. This test was recently described for the first time, by Ashworth et al., as the only test evaluating upper-quadrant isometric strength in a long lever position, presenting excellent reliability properties when performed to a group of elite adult rugby players [19]. Among the advantages of the test, the appropriateness to test positions that reproduce the sport needs, such as the tackle position, has been reported. Consequently, it could be of relevance for other sports such as tennis, where strokes like the serve, forehand, and overhead smash replicate the positions that the ASH test evaluates.

However, to our knowledge, the properties of the ASH test have not been examined in a young athletic population, offering a window of opportunity to investigate if the psychometric properties of such a test could be equally promising in an adolescent athletic population, an age category with very little evidence in terms of the reliability of strength-testing protocols. Therefore, the primary aim of this study was to investigate the inter-rater, intra-rater, and test-retest reliability of the test and report preliminary reference data for a sample of adolescent tennis players. The secondary purpose was to examine possible correlations between the ASH test and the rotational strength of the shoulder as well as the Seated Single-Arm Shot-Put Test (SSASPT), two of the most common metrics of strength and performance, as an indication of concurrent validity.

## 2. Materials and Methods

### 2.1. Design and Setting

A sample of convenience, including athletes from a single tennis academy, was used for the purposes of this cross-sectional study. Based on the previous literature, a minimum of 19 participants were required according to a priori calculation in order to be able to detect a moderate reliability score of 0.7 with the significance level set at 0.05 and a power of 80% [17]. A total of 33 healthy adolescent amateur tennis athletes from Fthiotikos Tennis Club (Lamia, Greece) volunteered to participate in this study. Measurements were taken at the local facilities of the Tennis Club between April and June 2023. Explanations concerning the measurement procedures were provided to both adolescent athletes and their parents before signing the informed consent. The study protocol was approved by the Physiotherapy Department Internal Ethics Committee of University of Thessaly (328/17-3-2023). All the procedures were conducted according to the Declaration and the reporting for this observational study is in line with the STROBE guidelines.

### 2.2. Sample

A total of 17 male and 16 female athletes with a mean age of 12.6 ± 2 years, a mean height of 1.58 ± 0.13 m, and a mean weight of 52.4 ± 14.6 kg made up the 33-tennis player sample. In order to be included in the study, athletes had to be practicing tennis for a minimum of one year and two hours weekly. Exclusion from participation was ensured in the cases of those currently experiencing symptoms of pain at the upper extremities, cervical and thoracic spine, previous injuries of the aforementioned body regions, and systemic diseases. The upper limb used to serve in tennis was defined as the dominant one, according to which 93,94% of the athletes were right-handed. More details of the anthropometric characteristics of the sample are presented in Table 1.

### 2.3. Procedure

Two novice junior physiotherapists with fewer than two years of clinical experience following their bachelor’s degree completion at the University of Thessaly, Greece, were in charge of the measurements (G.V and G.S). Both of them received training from a senior physiotherapist (A.P) for the execution of the testing protocols and completed three sessions of pilot testing to familiarize themselves with the procedures. A third physiotherapist (M.P) had the role of receiving the demographic and anthropometric data and coordinating the whole process. Each of the athletes participated in two sessions in order to complete the testing protocol. During the first session, after the demographic and anthropometric parameters’ recording, a five-minute standardized warm-up took place. The ASH test was then performed by the first physiotherapist (G.V), followed by rotator strength (internal and external) testing and the SSASPT. Between the three testing procedures, a ten-minute break was given to the tennis athletes to avoid fatigue effects and enable them to restore maximal strength. During the second session, which was scheduled with a one-week interval, only the ASH was conducted by the two different physiotherapists (G.V and G.S.). This was carried out to assess both the inter-session reliability of the same rater (G.V) and the inter-rater reliability between the two novice physiotherapists. The testing sequence between sides was randomized for all three tests on each occasion, as was the order of the two raters during the second session.

#### 2.3.1. Athletic Shoulder Test

The ASH was performed based on the original description of the test by Ashworth et al. [19]. Standardized instructions were provided to participants for the testing process, which always started after two submaximal contractions, which enabled the athletes to familiarize themselves with the procedure. Five-second contractions were performed during which the maximal force was advised to be produced within the first two and maintained for three more, while recovery periods of twenty seconds were offered. Three measurements per position (I,Y,T) and limb were recorded. The K-Force plates (Kinvent, Montpellier, France) were used for the conduction of the ASH test, a device that has been proven reliable for the test itself according to the previous literature [20,21]. In order to maximize the repeatability of the measurement procedure, fixed points were marked on the plates in order for the positioning of the hand on the plate to be standardized despite the upper-limb length differences (Figure 1). We ensured this by marking the point at exactly half the distance of the plate’s length and placing on it the mid part of the palm (defined as the mid of the distance between the tip of the middle finger and the radiocarpal joint).

#### 2.3.2. Shoulder Rotator Strength Test

External and internal rotator muscle strength testing of the tennis athletes took place according to the protocol described by Cools et al. [17]. Three measurements by side were used for each of the internal and external strength assessments to record the results of isometric testing using a ‘make’ test procedure. Again, five-second contractions with maximal force production within the first two and maintenance for the following three were adopted, followed by a rest period of twenty seconds. The K-Force Muscle Controller Handheld Dynamometer (Kinvent, Montpellier, France) was used for the tests, which were performed in a seated position. Participants sat on a non-arm chair with their back supported and their feet flat on the ground at an opening of shoulder width. The upper limb to be examined was positioned at 90° of abduction and 90° of external rotation, resting on a table. The dynamometer was placed two centimeters below the styloid process of the ulna, either on the dorsal side for the assessment of external rotation (ER) or on the ventral side for internal rotation (IR). The assessor ensured maximal stabilization of the participant’s setup through their contralateral hand, upper arm, and trunk. This position has previously shown excellent reliability, with intraclass correlation coefficient scores greater than 0.93. Although it scored slightly lower than the prone and supine positions, it does not appear to be prone to systematic errors [17].

#### 2.3.3. Seated Single-Arm Shot-Put Test

The SSASPT was assessed with participants seated against a wall, resting their back and head, with their knees fully extended and their arm by their side. The elbow was flexed so that the hand could maintain the medicine ball at the front of the shoulder, at the same height as the top of the acromion [22,23]. A standardized command was provided to ‘push the ball as far possible’, clarifying that excessive wrist movement should be avoided. The medicine ball used for the test weighed 2 kg, as the 6lbs in the initial description of the test, which equals a weight of 2.72 kg, was not an available option and we decided to use the lower weight of 2 kg for our adolescent population. The 2 kg medicine ball has also previously been used as a weight option for the Seated Medicine Ball Throw Test [24]. A numbered floor tape was placed to facilitate the measurement procedure as the chalk that covered the medicine ball could leave an easily identifiable print on it. Two submaximal throws were performed for familiarization purposes, followed by three maximal attempts per limb with a one-minute resting period between each of the trials. The mean of the three repetitions was used for the analysis.

## 3. Statistical Analysis

Descriptive statistics were used to present the demographic and anthropometric data as means, standard deviations, and ranges (Table). Relative reliability was assessed by calculating the two-way random effect ICC using absolute agreement (95% confidence interval) [25]. Inter-rater reliability was assessed by comparing the measurements the two physiotherapists recorded during the second session. Intra-rater reliability of the ASH was evaluated on two occasions for the first rater (first and second session) and once for the second rater (during the second session). Intra-rater reliability was calculated for the measurements of rotational strength and SSASPT that were recorded by the first rater during the first session, too. Test-retest reliability for the ASH was assessed by comparing the measurements of the first rater from the first session with those of the second. The interpretation of the ICC scores was classified according to the previous literature, as follows: excellent reliability = values > 0.90; good reliability = values between 0.80 and 0.89; moderate reliability = values between 0.70 and 0.79; fair reliability = values between 0.4 and 0.7; and poor reliability = values < 0.40 [26]. In order to minimize the risk of having high ICC scores due to intersubject variability despite lacking consistency, absolute reliability was also assessed through the standard error of measurement (SEM) and minimal detectable change (MDC). The SEM was calculated as the square root of the within-subject mean squared error from the repeated-measures analysis of variance, while the MDC was calculated by multiplying 1.96 × √2 × SEM [27]. Correlations between the ASH and rotational strength as well as SSASPT were examined with the Pearson Correlation Coefficient (r). The r values were categorized as weak (<0.40), moderate (0.4–0.69), strong (0.70–0.89), and very strong (0.9–1) [28]. The Statistical Package for Social Sciences (SPSS, SPSS Inc., Chicago, IL, USA) version 29.0 was used for the analysis.

## 4. Results

The demographic and anthropometric data can be found in Table 1. The ASH test scores are presented as means and standard deviations for the whole sample, as well as per gender and age group. Force, normalized torque, and peak torque were calculated per limb and position (Appendix A). ER, IR, and SSASPT scores are also presented per gender and age group apart from the whole sample data, with both absolute force and normalized torque measurements included for the rotational strength measurements (Appendix B). The intra-rater reliability scores of the aforementioned tests were calculated for the repeated measurements of the first rater (Section B.1). The ER-to-IR ratio was estimated and can be found in Section B.2.

### 4.1. Intra-Rater Reliability

The scores reflecting the repeated measurements demonstrated excellent reliability for both raters (Table 2, Table 3 and Table 4). More specifically, the ICC scores of the first rater ranged between 0.947 and 0.986 among the different positions of the ASH, while the scores of the second rater were almost identical (ICC = 0.970–0.983). The absolute reliability scores also confirm this, since SEM received scores between 3.24 and 7.06 newton and the MDC did not overcome 19.42 newton. Between the testing positions, ‘I’, which was always tested first, consistently presented the lower scores among the three. An additional observation was that the scores were slightly higher during the second session, which could hide a possible learning effect. Repeated measures of the rotational strength testing and the SSASPT also revealed excellent reliability rates (Section B.1).

### 4.2. Test-Retest Reliability

The investigation of the test-retest reliability via the measurements performed by the same rater separated by a week interval also presented excellent reliability scores (Table 5). More specifically, the ICC index received scores between 0.924 and 0.970, the SEM between 3.33 and 6.47, and the MDC between 9.32 and 18.04. Among the testing positions, relative reliability and absolute reliability provided some contrary results, as SEM again indicated ‘I’ as the position with the highest error of measurement, despite the ICC not being the lowest among the positions. A possible systematic error of the I position that repeatedly provided a standard measurement result could be the reason behind that.

### 4.3. Inter-Rater Reliability

Inter-rater reliability was found to be excellent, as the ICC values’ spectrum was scored between 0.974 and 0.988. No SEM score surpassed 3.92, and MDCs were below 10 newton for all positions except for the ‘I’ position of the dominant arm. The standardization of the test’s setup and instructions proved to enable the two novice physiotherapists to obtain similar testing scores even with minor levels of familiarization. An analytical presentation of the inter-rater reliability data examined only during the second session is illustrated in Table 6.

### 4.4. Correlation Between ASH, Rotational Strength, and SSASPT

A statistically significant positive correlation (*p* < 0.001) was found between all the ASH test positions and each of the ER, IR, and SSASPT (Table 7). For the dominant arm, all three positions of the ASH were strongly correlated with both rotational strength and SSASPT, apart from position I, which correlated moderately with ER (r = 0.605). In the non-dominant arm, correlation scores were lower, with the ER correlating only moderately with all the ASH positions, as was the case with the IR with position ‘T’. The SSASPT again provided strong correlation scores against all ASH positions, although lower than in the dominant arm (r = 0.755–0.823).

## 5. Discussion

To our knowledge, this was the first study to assess the ASH in a non-adult population and one of the few to assess the reliability of strength testing in adolescent overhead athletes. The results of our study are more than promising as all forms of reliability (intra-rater, inter-rater, and test-retest) proved to be excellent. These results reflect the results of Ashworth et al., who initially described the test, investigating its properties on an adult sample of overhead athletes [19]. Although direct comparisons are not permitted due to the sample characteristics and some minor design differences, such as the 5-s force development and maintenance period we used and the fact that peak force values were not considered when estimating reliability but only mean values, a few observations can be made upon closer inspection. More specifically, in both studies, higher force values were found in the ‘I’ position, followed by ‘Y’ and ‘T’. Additionally, the ‘I’ position showed the highest SEM and MDC scores in both cases. The lack of randomization during the protocol application prevents us from making any direct conclusions about possible influencing factors, such as learning effects or inherent difficulties of the position itself. Finally, the ICC indices are almost identical, with the SEM and MDC scores remaining low. However, when viewed as a proportion of the force values produced, the percentages are higher in our study. This could be attributed to the absence of familiarization sessions for our sample, which consisted of amateur athletes that had never participated in strength-testing procedures before. This emphasizes the simplicity and standardization of the test setup and its ability to produce consistent measurements.

In exploring possible associations between the ASH and other commonly performed upper-extremity tests, such as rotational strength of the shoulder and SSASPT, moderate to strong correlations were found. Lower correlation scores were observed between ASH and ER compared to IR and SSASPT. There are several possible explanations for these findings. First, the posterosuperior rotator cuff, responsible for external rotation force production, is thought to act as a counterbalancing mechanism to prevent anterior translation of the humeral head in flexion positions. This may not reflect the anterior musculature, including the internal rotators, which are the primary force producers during such movements [29]. Second, rotational strength testing is a somewhat more ‘isolated’ procedure that measures the force-producing capacity of a specific muscle group, while the higher correlations with SSASPT could be attributed to the more generic approach of upper-quadrant performance testing. Finally, the non-dominant extremity in a highly specialized dominant-arm sport like tennis may be expected to show more fluctuation in performance testing, which could explain the generally lower correlations observed on that side [30].

Apart from the force values per position and arm, we provided data for normalized and peak torque. The only study to date that has provided similar data for the ASH test was conducted by Olds et al., but this was on an adult, non-athletic population, which prevents any direct comparisons with our results [21]. In addition to the total sample scores for these variables, data were also provided by gender and age group. Male scores were higher in all positions for force, normalized torque and peak torque, except in the ‘T’ position, where females presented higher normalized values. Additionally, scores seemed to increase with age, which aligns with findings in the dynamometry literature for adolescent samples [31,32].

Intra-rater reliability scores for ER, IR, and SSASPT were found to be excellent, with ICC scores similar to those reported in the literature [17,22]. When divided by gender, males scored higher than females for all ER, IR, and SSASPT, except for normalized ER scores for both arms. In the age-group division, the 14–17-year-old group scored higher across all variables. The ER-to-IR ratios were higher for females than for males and for the younger age group (10–13) compared to the older group (14–17). Maturation could play a role here, as biological age is often considered more relevant than chronological age in adolescent athletes [4]. The non-linearity of strength development among adolescents across different muscle groups has been previously reported [33]. Compared to tennis data, similar ER-to-IR ratios have been reported by Cools et al. when looking at their sport discipline data for males, though they are lower for females [18]. In our study, the dominant arm provided higher ratios, which contrasts with their findings. In a separate study, where elite tennis players of similar ages were assessed with an isokinetic dynamometer, ratios were found to be higher than those in our study [34]. However, they used eccentric external rotation and concentric internal rotation at different angular velocities for calculating the ratios.

A potential limitation of the study design could be the uneven measurements between the two sessions. The implementation of two additional tests after performing the ASH just once during the first session could result in different fatigue levels compared to the second session, where only the ASH test was performed twice. Previous research has indicated that fatigue induced during a session could alter the repeated strength measurements [35]. However, our study did not follow a fatiguing protocol on purpose, so it is not that likely that fatigue played such a major role. In addition, learning effects and consistency during the second session’s measurements could have been maximized. In terms of generalizability of the findings to the full spectrum of adolescent athletes, apart from the small sample size, another limitation of our study was the age distribution, as the younger subgroup consisted of nearly twice the number of the athletes included in the 14–17-year-old age group. Furthermore, the convenience sample and the amateur level of the athletes may limit the findings when applied to tennis players of a similar age but at different competitive levels. Finally, the athletes were not familiarized with the testing protocol in our study, which could theoretically have improved reliability even further [19,20]. However, as highlighted by the previous literature, its relevance could be questionable when findings are considered for clinically relevant purposes rather than just for statistical significance [21].

## 6. Conclusions

Taking into consideration its excellent reliability rates, the ASH test protocol offers an upper-extremity assessment opportunity in adolescent athletes using easily applicable, portable, relatively cheap equipment, which can be performed even by novice physiotherapists with minor familiarization with the procedures. Future studies in both adolescent and adult populations of different sport disciplines will enhance our understanding of overhead athletes’ ASH performance either as a monitor tool for healthy populations or as part of the return to sport continuum.

## Figures and Tables

**Figure 1 jcm-14-01146-f001:**
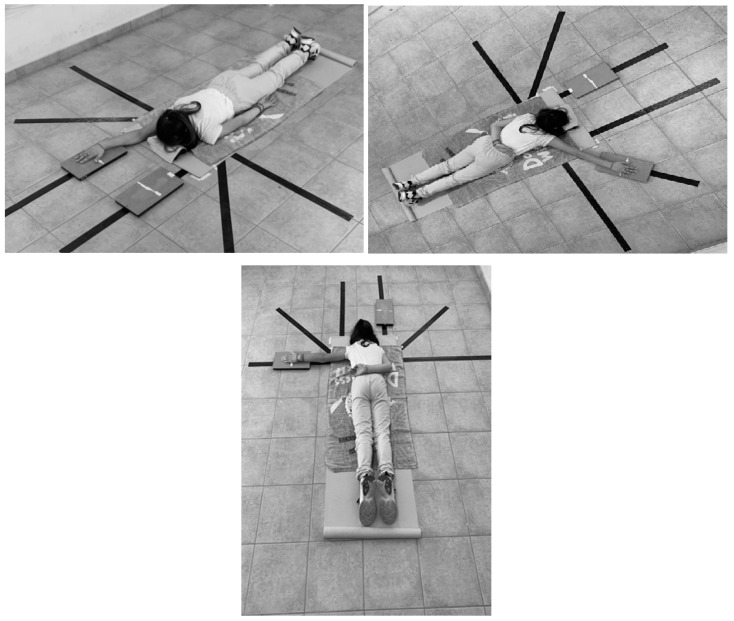
The I, Y, and T positions of the ASH test.

**Table 1 jcm-14-01146-t001:** Demographic and anthropometric data.

	Mean	Standard Deviation		Range
Age (years)	12.6	2		10–17
Weight (kg)	52.4	14.6		25.5–86.2
Height (m)	1.58	0.14		1.32–1.88
Right-Arm Length (cm)	53.1	5		43–65
Left-Arm Length (cm)	52.9	5		42.5–64
Tennis Participation (years)	5.1	2.9		1–9
Training Frequency (hours/week)	3.5	1.8		2–7
	Right		Left	
Dominant Arm (Number/%)	31 (93.9%)		2 (6.1%)	
	Male		Female	
Gender (Number/%)	17 (51.5%)		16 (48.5%)	
	10–13		14–17	
Age Group (Number/%)	21 (63.6%)		12 (36.4%)	

**Table 2 jcm-14-01146-t002:** Intra-rater reliability relative and absolute scores (Rater 1–1st session).

	Test Position	ICC	SEM	MDC
Dominant Arm	I	0.947 (0.856–0.974)	7.06	19.42
	Y	0.968 (0.933–0.984)	4.12	11.47
	T	0.977 (0.957–0.988)	3.43	9.51
Non-Dominant Arm	I	0.974 (0.953–0.987)	4.61	12.75
	Y	0.970 (0.953–0.987)	3.63	10.10
	T	0.966 (0.939–0.982)	3.82	10.69

**Table 3 jcm-14-01146-t003:** Intra-rater reliability relative and absolute scores (Rater 1—2nd session).

	Test Position	ICC	SEM	MDC
Dominant Arm	I	0.985 (0.971–0.993)	4.22	11.77
	Y	0.986 (0.975–0.993)	3.24	9.02
	T	0.977 (0.959–0.988)	3.63	10.10
Non-Dominant Arm	I	0.985 (0.972–0.992)	4.31	11.96
	Y	0.981 (0.966–0.990)	3.53	9.81
	T	0.981 (0.966–0.990)	3.24	8.82

**Table 4 jcm-14-01146-t004:** Intra-rater reliability relative and absolute scores (Rater 2).

	Test Position	ICC	SEM	MDC
Dominant Arm	I	0.981 (0.965–0.990)	5.19	14.42
	Y	0.982 (0.966–0.991)	3.43	9.41
	T	0.983 (0.969–0.991)	3.24	9.02
Non-Dominant Arm	I	0.982 (0.968–0.990)	4.71	11.96
	Y	0.970 (0.946–0.984)	4.02	11.08
	T	0.971 (0.949–0.985)	3.89	10.59

**Table 5 jcm-14-01146-t005:** Test-retest reliability relative and absolute scores (Rater 1—Between sessions measurements).

	Test Position	ICC	SEM	MDC
Dominant Arm	I	0.949 (0.898–0.975)	6.47	18.04
	Y	0.963 (0.925–0.981)	4.12	11.53
	T	0.924 (0.845–0.962)	5.29	14.71
Non-Dominant Arm	I	0.970 (0.939–0.985)	4.71	13.04
	Y	0.967 (0.934–0.984)	3.53	9.71
	T	0.959 (0.934–0.984)	3.33	9.32

**Table 6 jcm-14-01146-t006:** Inter-rater reliability relative and absolute scores.

	Test Position	ICC	SEM	MDC
Dominant Arm	I	0.983 (0.966–0.992)	3.92	10.98
	Y	0.982(0.963–0.991)	3.04	8.34
	T	0.974 (0.946–0.987)	3.24	9.02
Non-Dominant Arm	I	0.988 (0.975–0.994)	3.14	8.63
	Y	0.977 (0.953–0.988)	2.94	8.34
	T	0.978 (0.956–0.989)	2.74	7.55

**Table 7 jcm-14-01146-t007:** Correlations between ASH and rotational strength and SSASPT per side.

	Test Position	SSASPT (D)	ER (D)	IR (D)	SSASPT (ND)	ER (ND)	IR (ND)
Dominant Arm	I	0.808	0.605	0.772	-	-	-
	Y	0.856	0.742	0.867	-	-	-
	T	0.824	0.712	0.820	-	-	-
Non-Dominant Arm	I	-	-	-	0.755	0.584	0.771
	Y	-	-	-	0.823	0.631	0.752
	T	-	-	-	0.758	0.645	0.693

Force—Newton; Normalized Torque—Force/body mass (N/Kg); Peak Torque—Force × Arm Length (Nm).

## Data Availability

Data are available upon reasonable request from the corresponding author due to the adolescent population included.

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
