# Peer review of "Psychometric Properties of the Athletic Shoulder Test in Adolescent Tennis Players"

_jcm, 2025, doi:10.3390/jcm14041146_

Round 1
Reviewer 1 Report
Comments and Suggestions for Authors
ABSTRACT
Include SEM and MDC values
Replace vague mentions like "highly recommended" "supports ASH as highly reliable in adolescents"
MANUSCRIPT
Clarify that Schwank et al corresponds to the Bern Consensus statement
add state the missing data in adolescent-specific ASH validation to strengthen rationale
Cite the specific study used for a priori sample size calculation (referenced as previous literature).
Detail training duration/standardization for physiotherapists to allow protocol reproducibility
Add units to Table 1
Standardize decimal separators and hyphen consistency in ICC ranges
how protocol differences may limit comparability with Ashworth et al?
Expand on how uneven age distribution affects generalizability beyond stating it as a limitation.
Acknowledge the limitation of focusing on amateur athletes
qualify recommendations for elite adolescents.
Separate ASH (Athletic Shoulder Test) and ER (External Rotation).
Avoid combined terms like ASH ER
Author Response
Dear reviewer,
Thank you very much for your time spent reviewing this manuscript and the accurate comments in order to enhance its quality. Changes have been made accordingly. Pending some clarifications from editorial office to fully proceed with them.
Reviewer 2 Report
Comments and Suggestions for Authors
Thank you for this well-written manuscript about the feasibility of the ASH test in young people. It was interesting for me, but I have some suggestions.
Abstract: provides a clear overview of the study, outlining the background, methods, key findings, and conclusions.
Introduction: provides a detailed background on the significance of shoulder strength in tennis and the rationale for evaluating the ASH test.
Materials and Methods: well-structured and detailed description of the procedures, including intra-rater and inter-rater reliability assessments. Ethical approval and adherence to STROBE guidelines are appropriately noted.
However, there is no justification for using specific ICC thresholds and highlight the fact that there is a reduced generalizability as only a single tennis club with amateur level athletes is included. Please discuss this in more detail..
Results: the applied scores are reported comprehensively for all test positions and outcomes. Tables and figures effectively summarize the findings. However, results are overly focused on reliability metrics and lack broader interpretation.
Discussion: effectively compares the study findings with prior research, including Ashworth et al. and Olds et al. However, limitations are acknowledged but not critically analyzed, especially regarding the small sample size, and unequal age group distribution.
Conclusion: emphasizes the clinical utility of the ASH test.
General suggestion: After small adjustments and improvements as mentioned above, this manuscript should be able to meet the high requirements to be published in your journal.
sincerely
Author Response
Dear reviewer,
Thank you very much for your time spent reviewing this manuscript and the accurate comments in order to enhance its quality. Please find below the answers and the highlighted corrections on text:
Comment 1
Materials and Methods:well-structured and detailed description of the procedures, including intra-rater and inter-rater reliability assessments. Ethical approval and adherence to STROBE guidelines are appropriately noted.
However, there is no justification for using specific ICC thresholds and highlight the fact that there is a reduced generalizability as only a single tennis club with amateur level athletes is included. Please discuss this in more detail.
Response: Thank you for pinpointing this part. We followed previous research on strength testing reliability protocols, where they used the exact number as an ICC threshold (Cools et al, 2014 / in-text reference 17). A main reason on that was the fact that through the years, the low reliability scores of the upper extremity strength testing protocols were always considered to be an issue, thus we wanted an adequate sample size to reinforce the possible findings of a high index of relative reliability as the ICC. In terms of the sample characteristics, specifications have been done lines 80 & 85 as well as in the discussion section where the generalizability is more critically interpreted.
Comment 2
Results:the applied scores are reported comprehensively for all test positions and outcomes. Tables and figures effectively summarize the findings. However, results are overly focused on reliability metrics and lack broader interpretation.
Response: We deeply appreciate your feedback on this. We attempted to add a few points as a short comment / in-section interpretation (lines: 266-267 & 276-279), while during the first two paragraphs of discussion we have focused more on interpretation of these reliability metrics.
Comment 3
Discussion:effectively compares the study findings with prior research, including Ashworth et al. and Olds et al. However, limitations are acknowledged but not critically analyzed, especially regarding the small sample size, and unequal age group distribution.
Response: Thank you for highlighting the need for a critical analysis on this. Via minor changes, we tried to further include this in the discussion session, especially in lines 350-353, 356-359 & 367-372.
Reviewer 3 Report
Comments and Suggestions for Authors
Dear authors,
First of all thank you for the invitation to review your study “Psychometric properties of the Athletic Shoulder Test in adolescent tennis players” Please find some specific comments below.
ABSTRACT
- please revise the abstract and make it more refine and accretive with more statistical results.
INTRODUCTION
- The introduction benefit from description of the musculoskeletal problem related to overhead sport and their burden
- Please delineate more precisely the specific research gap that the study aims to fill
METHOD
- 2.3 Procedure: Please specify if the three physiotherapist had the same year of expertise and level of education.
RESULT
- The quality of your tables should be improved, please use only three lines
- The quality and the position of the figure should be revise
DISCUSSION
- Your discussion would benefit from a comparison with others study involved in other overhead sports and their assessment. For example, the fatigue play a pivot role in shoulder injuries especially for over-head sport. The activation of several muscles might contribute to fatigue after overhead sport. Fatigue may affects shoulder’s strength, proprioception, and range of motion, representing possible risk factors for overuse shoulder injury. Please take in consideration this article in our discussion section:
Buoite Stella A, Cargnel A, Raffini A, Mazzari L, Martini M, Ajčević M, Accardo A, Deodato M, Murena L. Shoulder Tensiomyography and Isometric Strength in Swimmers Before and After a Fatiguing Protocol. J Athl Train. 2024 Jul 1;59(7):738-744. doi: 10.4085/1062-6050-0265.23. PMID: 38014804; PMCID: PMC11277270.
- The main limitation of your study is the sample size that does not allow a gender stratification. In fact, studies reported several sex-based differences in muscles characteristics, moreover it seems that evaluation and management of shoulder and elbow sports injuries are different in female athletes. Please take in to consideration this article:
Wessel LE, Eliasberg CD, Bowen E, Sutton KM. Shoulder and elbow pathology in the female athlete: sex-specific considerations. J Shoulder Elbow Surg. 2021 May;30(5):977-985. doi: 10.1016/j.jse.2020.10.020. Epub 2020 Nov 18. PMI
Author Response
Dear reviewer,
Thank you very much for your time spent reviewing this manuscript and the accurate comments in order to enhance its quality. Please find below the answers and the highlighted corrections on text:
Comment 1
ABSTRACT
- please revise the abstract and make it more refine and accretive with more statistical results.
Response: Thank you for pinpointing this needed improvement with your comment. Changes have been made accordingly and the results section has been enriched with more statistical results as indicated.
Comment 2
INTRODUCTION
- The introduction benefit from description of the musculoskeletal problem related to overhead sport and their burden
- Please delineate more precisely the specific research gap that the study aims to fill
Response: We deeply appreciate your feedback on this. Relevant content was added and can been found in lines 50-53 and 80-83.
Comment 3
METHOD
- 2.3 Procedure: Please specify if the three physiotherapist had the same year of expertise and level of education.
Response: Thank you for highlighting the need for a more precise description on this. Clarification for the level of education and clinical experience was added and can be found in lines 118-120.
Comment 4
RESULT
- The quality of your tables should be improved, please use only three lines
- The quality and the position of the figure should be revise
Response: Thank for your comment on that. The quality and position of the figure were modified. In was not completely understood what the comment of three lines was exactly about. Our result tables were designed in accordance with previous original research were they provided the data of the ASH test analytically (Ashworth et al) and are quite similar to ones of Królikowska, et al and Olds, et al (in text references 19, 20 & 21).
Comment 5
DISCUSSION
- Your discussion would benefit from a comparison with others study involved in other overhead sports and their assessment. For example, the fatigue play a pivot role in shoulder injuries especially for over-head sport. The activation of several muscles might contribute to fatigue after overhead sport. Fatigue may affects shoulder’s strength, proprioception, and range of motion, representing possible risk factors for overuse shoulder injury.
- The main limitation of your study is the sample size that does not allow a gender stratification. In fact, studies reported several sex-based differences in muscles characteristics, moreover it seems that evaluation and management of shoulder and elbow sports injuries are different in female athletes.
Response: Your comment is very accurate and was one of our considerations. However, the design of the study and the procedures followed did not aim to cause a fatigue state and this is why did not focus that much in the discussion on this. The suggestion of your article was more than helpful though, so we did add a critical comment on the possible effects of fatigue in our study which can be found in lines 383-386. Finally, we acknowledge your really accurate comment on sex-based differences. This is why our main was not pinpoint this differences and we did not proceed with any statistical analysis on that. Nevertheless, as the literature is more than limited on providing pure values in athletic populations, especially non-adult, we followed the path of Cools et al (in-text reference 18) and provided a descriptive analysis of the pure values per gender and per age-group. In their study, when divided per gender and age group, there were also groups with as low as 19 athletes, while most groups included between 25 and 35 athletes. Thus, the generalizability of our findings may be limited, but it could serve as a precursor to future studies.
Reviewer 4 Report
Comments and Suggestions for Authors
Dear Authors,
I was pleased to review the paper entitled " Psychometric properties of the Athletic Shoulder Test in adolescent tennis players" -
- MDPI –
The present paper is very interesting, it focuses on a relevant clinical scenario, for orthopedics, potentially influencing the surgical and clinical practice for the management of tennis players.
Therefore, it is my opinion that the content is original, current, and relevant.
Thus, there are some minor remarks:
- Title: The title gives a fine idea of the topic to be covered.
- Abstract: please better explain “with measures of rotational shoulder strength and upper extremity performance were examined”.
- Introduction: Correct the layout of the manuscript by adhering to editorial standards.
Introduce better the role of this test in the current literature. Why do you think it should be better than other tests already present?
- Method: Better detail the inclusion and exclusion criteria. How many times were patients evaluated? Were there any other follow-ups before, during and after training?
The two images on page three need a new layout. They were cut off in the manuscript uploaded.
Also add images for the other two tests used as controls.
Start the statistics part, describing how you express the data e.g. mean and standard deviation or number and percentage.
- Results: Add description of acronyms under tables.
- Discussion: Compare your results more thoroughly with recent literature.
Using some inertial sensors routinely could better predict the evaluation of this test. You can add it as a future development in your research (you could add from doi: 10.5312/WJO.V12.I12.991).
Better describe the limitations, strengths, and possible future implications of your research
I recommend adding figures for all three pathologies selected in the analysis.
The paper generally is well written and needs only minor changes.
Author Response

(The authors gave the same response as above.)

Round 2
Reviewer 3 Report
Comments and Suggestions for Authors
Congratulation for the revision work done